# Potential mitigating role of ivermectin on the spread of *Chlamydia trachomatis* by *Musca sorbens*

Richard Selby[1]*, Anita Jeyam[1], Andrew Tate[1], Fikreab Kebede[2], Philip Downs[1]

1 Sightsavers international, 35 Perrymount Road, Haywards Heath, West Sussex, United Kingdom,
2 Ministry of Health, Addis Ababa, Ethiopia

* rselby@sightsavers.org

**Data Availability Statement:** Data used in this study is available from the ESPEN platform https://espen.afro.who.int/countries/ethiopia.

## Abstract

Trachoma is the world's most frequent cause of blindness from an infectious agent. The disease caused by infection is associated with lack of access to sanitation and low hygiene standards. Trachoma is controlled through the Surgery, Antibiotics, Facial cleanliness, and Environmental improvement (SAFE) strategy, which delivers azithromycin (AZM) mass drug administration (MDA) in endemic areas. The putative vector *Musca sorbens* principally reproduce in human faecal matter left in the environment due to open defecation. Ivermectin (IVM) is on the WHO's essential medicines list and is administered as preventative chemotherapy against two neglected tropical diseases (NTDs)—onchocerciasis, as an annual or bi-annual treatment, and lymphatic filariasis, as an annual treatment in combination with albendazole. Ivermectin has a known inhibitive effect on insects that reproduce in dung. To assess if IVM could be a viable vector control tool against *M. sorbens*, this study evaluates existing data from trachoma, onchocerciasis and lymphatic filariasis mass drug administration (MDA) operations in Ethiopia. Persistent and recrudescent trachoma in evaluation units (EUs) were examined for whether AZM MDA in EUs was accompanied by IVM MDA, and whether co-administration was associated with greater likelihood of trachoma control. Results show an association suggesting that EUs that received both IVM and AZM MDA benefit from improved control of trachoma in persistent or recrudescent areas, when compared to EUs that received AZM MDA. This initial investigation supports the potential for ivermectin's use to support SAFE. Findings warrant further work to validate ivermectin's impact on *M. sorbens* reproduction through controlled lab and field-based studies.

## Author summary

Trachoma is the world's most frequent cause of blindness from an infectious agent. Trachoma is controlled through the Surgery, Antibiotics, Facial cleanliness, and Environmental improvement (SAFE) strategy, which delivers azithromycin (AZM) mass drug administration (MDA) in endemic areas. The putative vector *Musca sorbens* principally reproduce in human faecal matter left in the environment due to open defecation.

**Funding:** Research presented in this paper has been undertaken with no specific funding. Sightsavers International is direct employer of authors (RS, AJ, AT and PD) and supported the time allocation for analysis of data used in this study. Other than salary support for the listed authors employed at Sightsavers international, the organisation has played no role in study design, data collection and analysis, decision to publish or preparation of the manuscript.

**Competing interests:** The authors have declared that no competing interests exist.

Ivermectin (IVM) is on the WHO's essential medicines list, administered against two neglected tropical diseases (NTDs)–onchocerciasis and lymphatic filariasis. Ivermectin has an inhibitive effect on insects reproducing in dung.

To assess if IVM could be a viable vector control tool against *M. sorbens*, this study evaluates existing data from trachoma, onchocerciasis and lymphatic filariasis mass drug administration (MDA) operations in Ethiopia. Persistent and recrudescent trachoma in evaluation units (EUs) were examined for whether AZM MDA in EUs was accompanied by IVM MDA, and whether co-administration could have controlled the putative vector, unintentionally enhancing trachoma control.

Results suggest that EUs receiving both IVM and AZM MDA had improved control of trachoma, when compared to EUs that have only received AZM MDA.

This initial investigation supports conducting further work into ivermectin's impact on *M. sorbens* reproduction through controlled lab and field-based studies.

## Introduction

Trachoma is the world's leading cause of blindness resulting from an infectious agent [1]. If left untreated trachoma leads to damage to the infected person's eyes. This damage culminates in blindness, which is avoidable with intervention [2]. Trachoma is strongly associated with poverty along with a lack of access to sanitation [3,4]. The disease is prevalent throughout the tropics, with areas of Africa, Asia, the Middle East, Western Pacific and Latin America affected [5]. The causative bacterium *Chlamydia trachomatis* can spread between people by direct contact, fomites or by face seeking flies which mechanically transmit bacteria [6].

Of the face seeking flies, *Musca sorbens* is the putative principal vector responsible for *C. trachomatis* mechanical transmission [6]. *M. sorbens* are dung breeding flies that most commonly reproduce in human faeces that have been defecated in the open environment [7]. *M. sorbens* feed on sweat, mucosal secretions (particularly ocular and nasal), lesions, wounds and ulcers of the skin [8]. Female *M. sorbens* obtain the protein needed for egg production directly from human facial secretions [9]. These factors combine to form a prime mechanism to transmit eye infections, including trachoma. Studies conducted in Gambia demonstrated that reducing fly populations through insecticide spraying reduces trachoma prevalence by 56% [6].

Blindness from trachoma can be avoided through surgical intervention on infected individuals to correct the blinding stage (trachomatous trichiasis) [2]. Control efforts encompass mass drug administration (MDA) with azithromycin (AZM) [10] coupled with improving water, sanitation and hygiene (WASH) [11,12]. The combination of surgery, MDA and WASH contribute toward the SAFE strategy (Surgery, Antibiotics, Facial cleanliness, and Environmental improvement) to eliminate trachoma as a public health problem, and the transmission of *C. trachomatis* [13].

Ivermectin's impact on dung breeding flies (*Musca nevilli*) has been investigated through treating cattle with sub-cutaneous injections of ivermectin (IVM) and letting flies oviposit on their dung [14]. Results showed reduced fly emergence from dung of IVM treated animals, lasting up to seven weeks post treatment. There was reduced fecundity amongst flies that successfully developed in dung of IVM treated animals [14]. Reproductive impact of IVM has been recorded for flies along with dung breeding beetles [15].

In addition, research into the acceptability of IVM MDA in Ghana, highlighted that communities had witnessed and appreciated IVM's impact on dung breeding flies. Qualitative data collection revealed that communities had noted *"Maggots in the public toilet died and that*

*further convinced us of the efficacy of the drug"* following IVM MDA. Existing work documents that IVM can be co-administered with other drugs including AZM in the context of NTD control [16,17] including in Ethiopia [18].

This led to initial questioning of whether IVM could provide a mechanism for *M. sorbens* control by effectively toxifying environmental human faeces? Would any IVM mediated impact upon *M. sorbens* populations affect trachoma, trachomatous inflammation–follicular (TF) persistence or recrudescence?

To investigate any potential impact of IVM on trachoma and TF, we conducted the following study focusing on Ethiopia, which has the highest burden of active trachoma and greatest recorded incidence of persistent and recrudescent TF globally [1,19]. Along with endemic areas for both onchocerciasis (OV) and lymphatic filariasis (LF), where MDA with IVM is conducted annually treating all individuals over 5 years of age [20,21]. As of 2020, sixteen percent of the Ethiopia population practice open defecation [22].

This document describes analysis conducted to investigate any previously unexplored association between IVM MDA and trachoma prevalence in Ethiopia. Authors highlight this study has been exploratory only, achieved by combining different data sources post-hoc to examine associations. This study aims to build a base of knowledge to guide further investigation.

## Methods

For this secondary analysis, data were extracted for IVM and AZM administration throughout Ethiopia. Data concerning onchocerciasis and LF were obtained from the Expanded Special Project for Elimination of Neglected Tropical Diseases (ESPEN) platform between 2014 and 2020. Data for trachoma were also obtained through ESPEN, and using the GeoConnect administrative coding system. Trachoma data were evaluated between 2007 and 2020. Date ranges were selected to ensure completion and accuracy of data. From these datasets we identified locations based on two classifications:

1) Where onchocerciasis or lymphatic filariasis were co-endemic with trachoma (and received multiple rounds of IVM and AZM).

2) Where only trachoma is known, and AZM has been administered without IVM.

Trachoma data were summarized to one record per Evaluation Unit (EU). For each EU, information was gathered from the baseline survey up to the most recent impact (or surveillance) survey that categorised the EUs as either persistent or recrudescent.

To conduct this analysis we have used the following working definitions, as approved by WHO [23]:

A EU with persistent trachoma is classified as an EU with at least two impact surveys at which trachomatous inflammation-follicular (TF) prevalence in one to nine year olds ($TF_{1-9}$) is $\geq$ 5%, without ever having had a $TF_{1-9} < 5\%$.

A EU with recrudescent trachoma has at least one surveillance survey at which $TF_{1-9}$ returns to $\geq$ 5%, and the current $TF_{1-9}$ prevalence is $\geq$ 5%.

To evaluate trachoma, the data set was examined using a predefined sequential criterion, see Fig 1, using one record per EU defined by a unique GeoConnect identifier (ID). These records contain information on survey types, years and category of $TF_{1-9}$ prevalence during each survey.

Using the outlined definitions to explore persistence and recrudescence of trachoma [23]:

Recrudescence was examined among all EUs that had at least one surveillance survey.

Persistence was examined among the EUs that had 2 or more impact surveys.

Data for trachoma was merged with available onchocerciasis and LF data (recorded by ESPEN IDs), using a key Masterfile, matching ESPEN IDs and GeoConnect IDs.

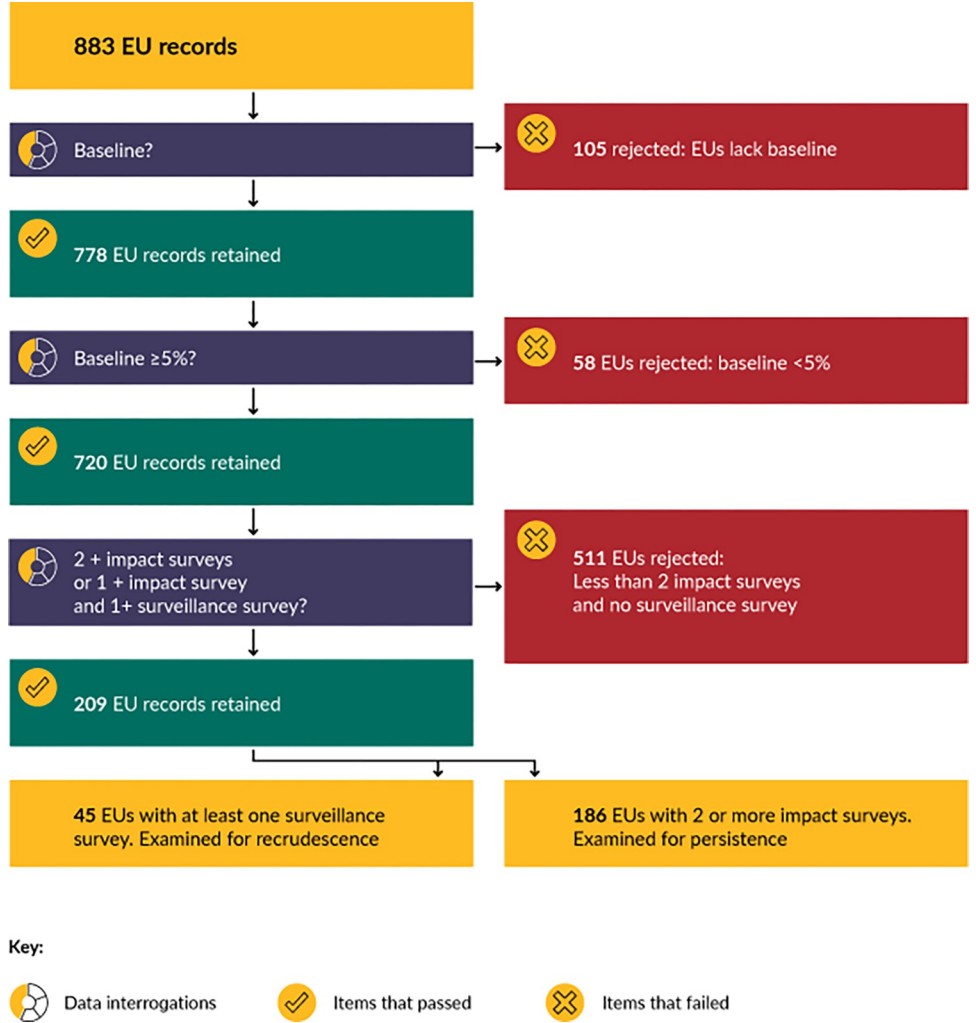

**Fig 1. Exclusion and inclusion process of Ethiopian EUs for evaluating TF persistence and recrudescence against IVM and AZM MDA.** Note, 22 of the retained EUs matched criteria used for both recrudescence and persistence analysis.

Within the onchocerciasis and LF data, EUs were examined for if they had received any IVM MDA (either on its own or in association with another drug).

There are known variables that can impact *M. sorbens* reproduction and survival [24]. To examine the influence of these variables on this investigation available geographic data have been collated for elevation [25–27], precipitation [28], landcover [29], population density [30], sanitation and water access [31]. These data were overlaid with the plots of EUs within our datasets for examination. Mean values for precipitation, elevation and population density per EU were calculated using zonal statistics in ArcGIS. For landcover, maximum coverage by land cover type (1km spatial resolution) per EU was utilised. Estimates for the proportion of people with access to improved sanitation and water were obtained from the ESPEN platform.

Statistical analyses were conducted at the EU level, using R v4.2.2. [32], using a significance level of 5%, no imputation of missing data was performed. Chi-square and fisher exact tests were used to examine the association between IVM and persistence/recrudescence. Within-sample quartiles were used to describe precipitation, elevation. Univariate logistic models were

**Table 1. Cross tabulation of trachoma recrudescent EUs and mass IVM administration.**

|  | Recrudescent: no | Recrudescent: yes | P (Fischer exact) |
|---|---|---|---|
| IVM MDA: No | 14 [38%] | 21 [62%] |  |
| IVM MDA: Yes | 10 [100%] | 0 | <0.01 |

used to explore associations between potential individual confounders and persistence of trachoma. A multivariable model with simultaneous variable entry was used to assess the association between IVM and persistence adjusting for all potential confounders explored in this study. We supplemented this with a data-driven variable selection approach, using a backward selection procedure based on the AIC criterion to define an optimal parsimonious model.

## Results

During data collation, EUs that could not be aligned from ESPEN into GeoConnect were discounted from our dataset prior to assessment commencing. This assessment commenced with 883 records for trachoma, one record mapped per EU, defined by a unique GeoConnect ID that contained information on the survey types, years of surveys and category of $TF_{1-9}$ prevalence.

EUs were sequentially excluded due to lacking a baseline survey, retaining only EUs with baseline surveys. These EUs were then assessed to retain only those with a prevalence of $TF_{1-9}$ ≥5% at baseline survey. Finally, to explore persistence and recrudescence, we only retained the 209 EUs with either 2 or more impact surveys or at least one impact survey and one surveillance survey.

We examined recrudescence among the 45 EUs that had at least one surveillance survey. Persistence was examined in the 186 EUs that were retained. Steps taken during the process are depicted in Fig 1.

As an initial exploration, simple cross-tabulations are presented between trachoma recrudescence and IVM MDA (Table 1) along with trachoma persistence and IVM distribution (Table 2).

Forty-four EUs that have received AZM MDA with at least one surveillance survey registered, were examined for trachoma recrudescence. Thirty-five of these EUs have not received IVM MDA, of which 13 did not fit the WHO criteria as being trachoma recrudescent and 21 meet classification as trachoma recrudescent. Among the ten EUs that have received both IVM MDA and AZM MDA, none of which met classification as TF recrudescence. There was a significant association between IVM MDA and recrudescence status (p<0.01).

In total 186 EUs had two or more impact surveys and were examined for trachoma persistence. In total 164 EUs had received only AZM MDA, of these 116 EUs were defined as trachoma persistent and 48 were not trachoma persistent. Twenty-two EUs had received both AZM and IVM. Of which 13 were classified as trachoma persistent and 9 were not trachoma persistent. There was a significant association between IVM MDA and persistence status (p<0.01).

Location of Ethiopian EUs are shown in Fig 2. Mapping EUs with and without IVM co-administration shows EUs with persistent and recrudescent trachoma have minimal overlap

**Table 2. Cross tabulation of trachoma persistent EUs and mass IVM administration.**

|  | Persistent: no | Persistent: yes | P (chi-square) |
|---|---|---|---|
| IVM MDA: No | 48 [29%] | 116 [71%] |  |
| IVM MDA: Yes | 13 [61%] | 9 [39%] | <0.01 |

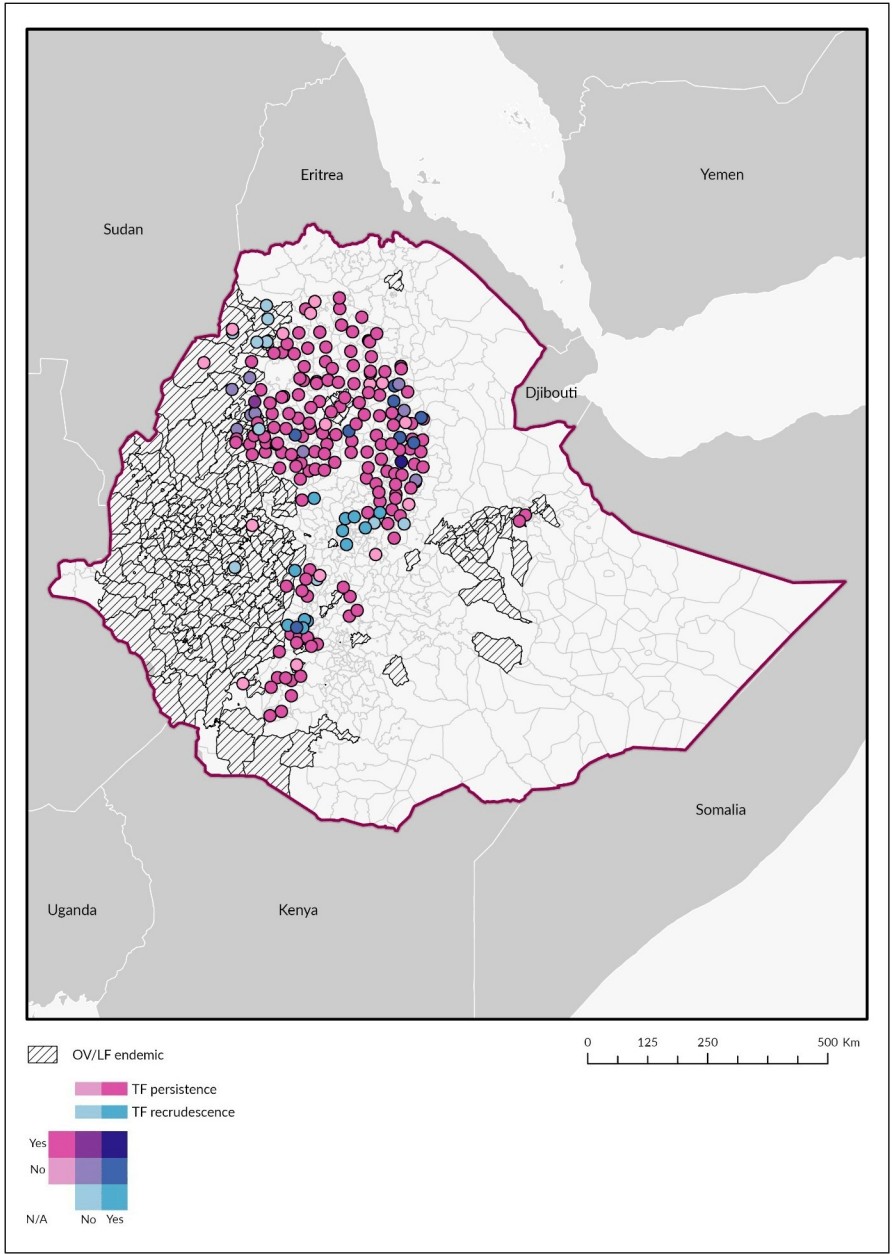

**Fig 2. Locations of Ivermectin MDA related to Trachoma persistence and recrudescence status in Ethiopia.**
Shapefile: https://espen.afro.who.int/tools-resources/cartography-database.

with IVM MDA. EUs that have received both AZM and IVM are less likely to have persistent or recrudescent trachoma among their population.

## Confounders

Variables that could influence trachoma transmission rates are shown in maps that make up Fig 3, with one panel per variable. Maps are overlaid with the trachoma data and IVM MDA areas (for OV and LF) as presented in Fig 2.

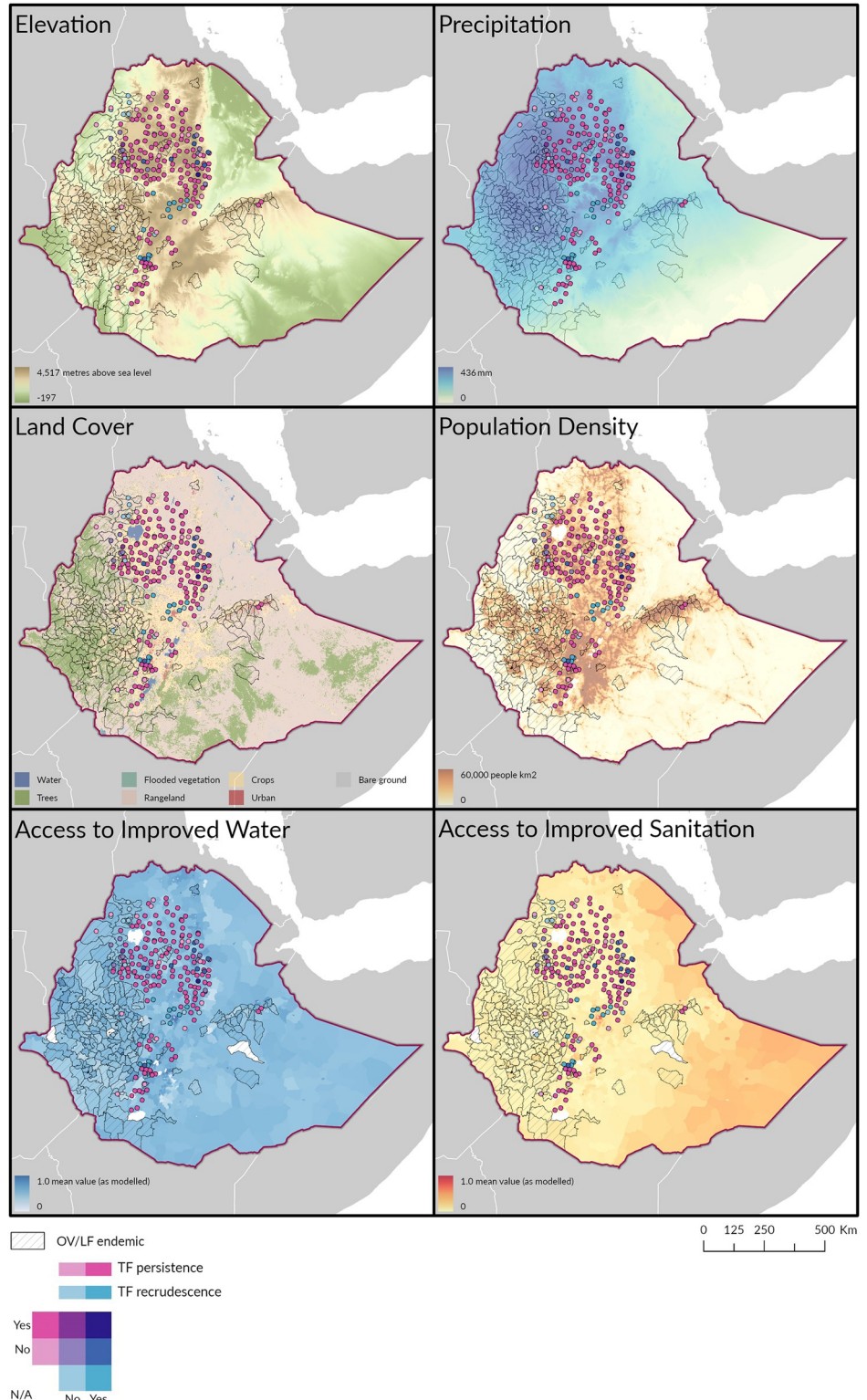

**Fig 3. Maps of influencing factors throughout EUs included in this study.** Clockwise from top left, elevation [27], precipitation, population density, access to improved sanitation, access to improved water and landcover. Shapefile: https://espen.afro.who.int/tools-resources/cartography-database.

Average precipitation, compiled by Fick and Hijmans [28] from August (peak rainfall in Ethiopia), indicates rainfall patterns are not vastly different between EUs receiving IVM MDA compared to trachoma associated EUs which met the criteria for inclusion in this study. Plotting landcover shows that EUs which are endemic for OV/LF where IVM MDA occurred were primarily classified as being forested with trees with some areas classified as crop, whereas the trachoma EUs are primarily classified as cropland and rangeland (grass and scrub). Population density throughout the OV/LF and trachoma EUs have a similar range, but it is notable that more IVM MDA EUs have lower categories of population density than the AZM MDA EUs. Comparison of access to improved water throughout the EUs included in this study does not reveal stark differences between trachoma EUs and OV/LF EUs. Access to improved sanitation is similar between the two classifications of EUs, with little improvement in the trachoma EUs that have also received IVM MDA.

## EU characteristics by trachoma—persistence and recrudescence status

Table 3 details the EU confounder characteristics by trachoma status—recrudescence and persistence.

Among EUs with recrudescent trachoma, there was a higher proportion of EUs with lower baseline TF prevalence when compared to non-recrudescent trachoma EUs. The number of AZM administration years was lower. Mean population density was lower. The mean EU elevation was higher. The proportion of EUs with mean proportion of people with improved water access being in the highest quartile was lower, similarly for improved access to sanitation and the proportion of EUs with majority cropland was higher.

Compared to non-persistent EUs, among persistent EUs, the proportion with highest category of baseline TF prevalence was higher, as was the proportion with highest number of AZM years, the mean population density was lower, as was the mean elevation. The proportion of EUs with mean proportion of people with improved water access in the highest quartile was lower, similarly for improved sanitation, whereas the proportion of EUs in the highest precipitation quartile was lower. Finally, the proportion of EUs with majority cropland was lower while those with majority rangeland was higher.

Statistical modelling was only conducted on trachoma-persistence since the sample size for recrudescence was small (n = 45) and there was quasi-complete separation in the data with none of the recrudescent EUs having received IVM, the outcomes are given in Table 4.

Univariate models show that EUs that had received IVM were less likely to present persistent trachoma (OR = 0.29 [0.11, 0.71]). As were those with a lower number of AZM years (OR = 0.07 [0.01, 0.30] for 0–5 AZM years compared to more than 10 years). Those with highest mean population density (OR = 0.22 [0.08,0.53] for those with mean 318 or more people/ km$^2$ versus those with mean<152), those with higher elevation levels (OR = 0.35 [0.14, 0.85] for those with mean elevation 1740m/100m$^2$ versus those with <1060m/100m$^2$). Finally, the EUs with maximum land cover being rangeland were more likely to present persistent trachoma than those with maximum cropland (OR = 4.10 [1.86, 10.06]). There was no evidence of statistically significant associations between trachoma persistence and baseline TF prevalence levels, water or sanitation access nor precipitation levels.

In the full multivariable model adjusting for all potential confounders available, the association between IVM and persistence status is not statistically significant (p = 0.15). However, the optimal model resulting from the backward selection process includes IVM, number of AZM years, population density and precipitation levels. Results from this parsimonious model show that EUs with IVM, with a lower number of AZM years, higher population density and higher levels of precipitation were less likely to present persistent trachoma.

**Table 3. EU characteristics by trachoma recrudescence and persistence status—descriptive statistics.**

| | | Recrudescence | | Persistence | |
|---|---|---|---|---|---|
| | | **Not recrudescent** | **Recrudescent** | **Not persistent** | **Persistent** |
| IVM occurrence between baseline and last survey for trachoma | No | 14 (58.3%) | 21 (100%) | 48 (78.7%) | 116 (92.8%) |
| | Yes | 10 (41.7%) | 0 (0%) | 13 (21.3%) | 9 (7.2%) |
| Baseline survey TF prevalence category | 10–29.9 | 8 (33.3%) | 18 (85.7%) | 28 (45.9%) | 61 (48.8%) |
| | 30–49.9 | 14 (58.3%) | 1 (4.8%) | 28 (45.9%) | 44 (35.2%) |
| | > = 50 | 2 (8.3%) | 2 (9.5%) | 5 (8.2%) | 20 (16.0%) |
| Number of AZM years between baseline and last survey for trachoma | 0 | 0 (0%) | 0 (0%) | 2 (3.3%) | 0 (0.0%) |
| | 1 to 5 | 9 (37.5%) | 12 (57.1%) | 3 (4.9%) | 4 (3.2%) |
| | 6 to 10 | 15 (62.5%) | 9 (42.9%) | 49 (80.3%) | 37 (29.6%) |
| | > 10 | 0 (0%) | 0 (0%) | 7 (11.5%) | 84 (67.2%) |
| Mean EU elevation—quartiles (m/100m$^2$) | <1060 | 8 (33.3%) | 3 (14.3%) | 12 (19.7%) | 33 (26.4%) |
| | 1060–1350 | 5 (20.8%) | 2 (9.5%) | 12 (19.7%) | 38 (30.4%) |
| | 1350–1740 | 4 (16.7%) | 8 (38.1%) | 17 (27.9%) | 35 (28.0%) |
| | > 1740 | 7 (29.2%) | 8 (38.1%) | 20 (32.8%) | 19 (15.2%) |
| Mean EU precipitation–quartiles (mm/Km$^2$) | <212 | 3 (12.5%) | 8 (38.1%) | 13 (21.3%) | 32 (25.8%) |
| | 212–257 | 6 (25.0%) | 5 (23.8%) | 11 (18.0%) | 35 (28.2%) |
| | 257–298 | 3 (12.5%) | 6 (28.6%) | 17 (27.9%) | 30 (24.2%) |
| | >298 | 12 (50.0%) | 2 (9.5%) | 20 (32.8%) | 27 (21.8%) |
| Maximum land coverage type | Cropland | 13 (54.2%) | 15 (71.4%) | 45 (73.8%) | 70 (56.0%) |
| | Rangeland | 6 (25.0%) | 4 (19.0%) | 8 (13.1%) | 51 (40.8%) |
| | Urban | 5 (20.8%) | 2 (9.5%) | 8 (13.1%) | 2 (1.6%) |
| | Water | 0 (0%) | 0 (0%) | 0 (0.0%) | 1 (0.8%) |
| | Trees | 0 (0%) | 0 (0%) | 0 (0.0%) | 1 (0.8%) |
| Mean EU population density quartiles (people/Km$^2$) | 0–152 | 6 (25.0%) | 2 (9.5%) | 10 (16.4%) | 37 (29.6%) |
| | 152–211 | 4 (16.7%) | 9 (42.9%) | 11 (18.0%) | 32 (25.6%) |
| | 211–318 | 3 (12.5%) | 3 (14.3%) | 15 (24.6%) | 36 (28.8%) |
| | > = 318 | 11 (45.8%) | 7 (33.3%) | 25 (41.0%) | 20 (16.0%) |
| Mean proportion of people with improved water access—quartiles | <0.63 | 4 (19.0%) | 5 (25.0%) | 15 (27.8%) | 38 (30.9%) |
| | 0.63–0.68 | 7 (33.3%) | 1 (5.0%) | 12 (22.2%) | 34 (27.6%) |
| | 0.68–0.72 | 2 (9.5%) | 9 (45.0%) | 10 (18.5%) | 32 (26.0%) |
| | >0.72 | 8 (38.1%) | 5 (25.0%) | 17 (31.5%) | 19 (15.4%) |
| Mean proportion of people with improved sanitation—quartiles | < .07 | 7 (33.3%) | 6 (30.0%) | 10 (18.5%) | 40 (32.5%) |
| | 0.07–0.09 | 6 (28.6%) | 6 (30.0%) | 19 (35.2%) | 37 (30.1%) |
| | 0.09–0.11 | 1 (4.8%) | 4 (20.0%) | 7 (13.0%) | 21 (17.1%) |
| | >0.11 | 7 (33.3%) | 4 (20.0%) | 18 (33.3%) | 25 (20.3%) |

## Discussion

This initial investigation suggests that MDA of IVM could be associated with lower odds of trachoma persistence and recrudescence. We hypothesise that IVM could be inadvertently delivering *M. sorbens* control by toxifying environmental human faeces, in turn this IVM mediated impact upon *M. sorbens* populations has supported SAFE strategy's control of trachoma (TF). These findings warrant expansion of research to investigate the impact of IVM on *M. sorbens*, trachoma transmission and any environmental impact of IVM contaminated open defecation by humans. Such research would deliver an evaluation for repurposing IVM to support trachoma campaigns in difficult foci.

**Table 4. Results of statistical modelling confounders.**

| | | Univariable logistic regression | | Multivariable model results–simultaneous variable entry | | Multivariable model results, backward selection results | |
|---|---|---|---|---|---|---|---|
| | | Odds Ratio [95% CI] | p | Odds Ratio [95% CI] | p | Odds Ratio [95% CI] | p |
| IVM occurrence between baseline and last survey for trachoma | No | ref | <0.01 | ref | 0.15 | ref | 0.03 |
| | Yes | 0.29 [0.11, 0.71] | | 0.34 [0.08, 1.46] | | 0.24 [0.06, 0.86] | |
| Baseline survey TF prevalence category | 10–29.9 | ref | 0.22 | ref | 0.14 | | |
| | 30–49.9 | 0.72 [0.37, 1.38] | | 0.30 [0.08, 1.09] | | | |
| | > = 50 | 1.84 [0.66, 5.96] | | 1.80 [0.27, 14.00] | | | |
| Number of AZM years between baseline and last survey for trachoma | 0 to 5 | 0.07 [0.01, 0.30] | <0.01 | 0.02 [0.00, 0.23] | <0.01 | 0.04 [0.01, 0.22] | <0.01 |
| | 6 to 10 | 0.06 [0.02, 0.14] | | 0.01 [0.00, 0.05] | | 0.02 [0.00, 0.06] | |
| | > 10 | ref | | ref | | ref | |
| Mean EU elevation—quartiles (m/100m$^2$) | <1060 | ref | 0.04 | ref | 0.45 | | |
| | 1060–1350 | 1.15 [0.45, 2.93] | | 2.06 [0.46, 9.88] | | | |
| | 1350–1740 | 0.75 [0.31, 1.79] | | 2.07 [0.44, 10.19] | | | |
| | >1740 | 0.35 [0.14, 0.85] | | 0.75 [0.12, 4.54] | | | |
| Mean EU precipitation–quartiles (mm/Km$^2$) | <212 | ref | 0.25 | ref | 0.04 | ref | <0.01 |
| | 212–257 | 1.29 [0.51, 3.34] | | 0.18 [0.03, 0.80] | | 0.30 [0.08, 1.05] | |
| | 257–298 | 0.72 [0.29, 1.72] | | 0.08 [0.01, 0.44] | | 0.07 [0.02, 0.28] | |
| | >298 | 0.55 [0.23, 1.29] | | 0.19 [0.03, 1.15] | | 0.08 [0.02, 0.31] | |
| Maximum land coverage type | Cropland | ref | <0.01 | ref | 0.58 | | |
| | Rangeland | 4.10 [1.86, 10.06] | | 0.58 [0.04, 7.21] | | | |
| | Urban, Water, Trees | 0.32 [0.08, 1.08] | | 2.04 [0.42, 10.32] | | | |
| Mean EU population density quartiles (people/Km$^2$) | 0–152 | ref | <0.01 | ref | 0.25 | ref | <0.01 |
| | 152–211 | 0.79 [0.29, 2.10] | | 0.28 [0.05, 1.38] | | 0.33 [0.08, 1.30] | |
| | 211–318 | 0.65 [0.25, 1.62] | | 0.19 [0.03, 1.12] | | 0.17 [0.04, 0.66] | |
| | > = 318 | 0.22 [0.08, 0.53] | | 0.14 [0.02, 0.96] | | 0.08 [0.02, 0.29] | |
| Mean proportion of people with improved water access—quartiles | <0.63 | ref | 0.11 | ref | 0.3 | | |
| | 0.63–0.68 | 1.12 [0.46, 2.76] | | 1.19 [0.29, 4.93] | | | |
| | 0.68–0.72 | 1.26 [0.50, 3.27] | | 4.14 [0.95, 20.84] | | | |
| | >0.72 | 0.44 [0.18, 1.06] | | 1.51 [0.39, 6.01] | | | |
| Mean proportion of people with improved sanitation—quartiles | < .07 | ref | 0.12 | ref | 0.7 | | |
| | 0.07–0.09 | 0.49 [0.19, 1.16] | | 0.58 [0.14 2.41] | | | |
| | 0.09–0.11 | 0.75 [0.25, 2.33] | | 1.15 [0.21, 6.39] | | | |
| | >0.11 | 0.35 [0.13, 0.86] | | 0.61 [0.14, 2.72] | | | |

For the analysis presented here, we did not have access to dated historical data on IVM for OV and LF pre-2014. However, the earliest year of last trachoma survey considered for EUs was 2014 and IVM is not anticipated to have long-term residual impact capable of affecting *M. sorbens* reproduction. The plasma half-life of IVM in humans is approximately 18 hours following oral administration. Orally administered IVM and its metabolites are excreted in human faeces for 12 days post treatment [33].

The locations where IVM has been administered do not overlap with ongoing persistent or recrudescent trachoma. We have made efforts to investigate whether the spatial segregation is due to other influencing factors. Evaluating elevation, precipitation, land cover, population density, water and sanitation access throughout the EUs for trachoma, onchocerciasis and lymphatic filariasis transmission. Of note, authors made efforts to plot findings of this study

alongside datasets for socio economic variables, but viable resolution for Ethiopia was not attainable for assessment.

It was expected that OV and LF endemic EUs would likely receive higher rainfall patterns than non-endemic EUs, as transmission of both depend on vectors which develop in aquatic environments. However, Fig 3 shows similar variations of rainfall throughout the investigated areas. This is relevant as rainfall accelerates the breakdown of faeces in the environment, removing the reproductive substrate for *M. sorbens.* which could have generated spatial differences in trachoma control. Plotting other factors that could influence trachoma transmissions indicates no stand-out difference in population density, access to improved sanitation or improved water access as presented. There is a viable difference in the landcover classifications throughout the OV and LF areas when compared to the trachoma endemic areas. But when considering EUs that had co-administration of AZM and IVM, the landcover includes both forest and cropland classifications present in these EUs that received IVM MDA, and trachoma reduced.

Variations in heavy rainfall could have driven differences in the abundance of environmental faecal matter and the speed of its decay. Thus, rainfall could impact *M. sorbens* reproduction between EUs with and without IVM MDA. While Fig 3 indicates this not to be the case. Authors acknowledge that Fig 3 uses the peak rainfall of August and other months may present variations that could influence *M. sorbens* reproduction. We have attempted to account for potential confounders that could affect the observed association between IVM MDA and enhanced trachoma control, using multivariable models. The model results are encouraging and highlight the need for future research to better understand the mechanisms at play behind the relationship between IVM, EU characteristics and trachoma so they are accurately reflected in the model building.

This research provides initial evidence that community wide IVM MDA may have a role in reducing trachoma transmission by delivering vector control. Future research into the impact on laboratory controlled and directly observed *M. sorbens* reproduction is vital. Laboratory research should be accompanied by case control investigations based in Ethiopia. Any field research should consider the introduction of IVM MDA in selected case EUs with both recrudescent or persistent trachoma with continuation of SAFE strategy including AZM MDA as currently managed.

## Limitations

Authors highlight and acknowledge limitations within the study presented in this paper, which has been conducted as an assessment of the initial concept's viability.

The analysed data is on an EU level only, repurposing data that had not been collected for the purpose of these analyses. We lack the granularity of individual-level data that would allow us to investigate socio-demographic and behavioural factors. However, population level effect on prevalence of TF is the principal outcome of interest while evaluating the potential of IVM for *M. sorbens* vector control. Authors highlight that some covariates used in our analysis have a degree of uncertainty, being themselves estimates derived from modelling, such as the WASH data for example.

As we have mentioned above and reiterate here, our analyses do not allow to infer a causal relationship between IVM MDA and lower odds of trachoma persistence or recrudescence and more research is warranted to understand the mechanisms driving this association.

## Conclusion

Our analysis of existing data suggests an association between IVM MDA and improved control of trachoma in persistent or recrudescent areas. This finding warrants further research to ascertain the potential of IVM MDA implementation for trachoma control.

Following this initial assessment authors will expand the investigation by commencing insectary-based evaluation of *M. sorbens* reproduction in IVM contaminated human faeces. Concurrently, impact of IVM on adult flies will be investigated, to assess if adult fly mortality increases after imbibing facial secretions from an IVM treated person, as observed in heamato-phagic ectoparasites [34,35].

## Acknowledgments

Authors wish to express their gratitude to the Ethiopian Ministry of Health, International Trachoma Initiative (ITI), ESPEN and GeoConnect platforms for making the data used in this study openly available. We thank Andrew Balchin for support with graphical design of Fig 1 for use in the paper.

## Author Contributions

**Conceptualization:** Richard Selby, Philip Downs.

**Data curation:** Anita Jeyam, Andrew Tate, Fikreab Kebede.

**Formal analysis:** Richard Selby, Anita Jeyam, Andrew Tate, Philip Downs.

**Investigation:** Richard Selby, Andrew Tate.

**Methodology:** Richard Selby, Anita Jeyam, Andrew Tate, Philip Downs.

**Project administration:** Richard Selby.

**Resources:** Andrew Tate.

**Supervision:** Richard Selby.

**Validation:** Anita Jeyam.

**Visualization:** Richard Selby, Fikreab Kebede.

**Writing – original draft:** Richard Selby.

**Writing – review & editing:** Anita Jeyam, Andrew Tate, Fikreab Kebede, Philip Downs.

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
