## [Decision Letter · Decision Letter 0]

10 Apr 2023

Dear Dr Selby,

Thank you very much for submitting your manuscript "Potential mitigating role of ivermectin on the spread of Chlamydia trachomatis by Musca sorbens." for consideration at PLOS Neglected Tropical Diseases. As with all papers reviewed by the journal, your manuscript was reviewed by members of the editorial board and by several independent reviewers. In light of the reviews (below this email), we would like to invite the resubmission of a significantly-revised version that takes into account the reviewers' comments. 

We cannot make any decision about publication until we have seen the revised manuscript and your response to the reviewers' comments. Your revised manuscript is also likely to be sent to reviewers for further evaluation.

Sincerely,

Susana Vaz Nery

Academic Editor

Álvaro Acosta-Serrano

Section Editor

Reviewer's Responses to Questions

**Key Review Criteria Required for Acceptance?**

**Methods**

-Are the objectives of the study clearly articulated with a clear testable hypothesis stated?

-Is the study design appropriate to address the stated objectives?

-Is the population clearly described and appropriate for the hypothesis being tested?

-Is the sample size sufficient to ensure adequate power to address the hypothesis being tested?

-Were correct statistical analysis used to support conclusions?

-Are there concerns about ethical or regulatory requirements being met?

Reviewer #1: Yes, yes, mostly, sample size was dictated by available data, partially, no (though please see comment about list of authors)

Reviewer #2: This paper reviews data on the prevalence of active trachoma in Ethiopia, and shows that evaluation units (EUs) which received mass treatment with ivermectin in addition to azithromycin were less likely to have persistent or recrudescent active trachoma than those which received azithromycin alone. The authors speculate that this may be due to the fact that ivermectin in human faeces inhibits the breeding of Musca sorbens, a species of fly that has been implicated in the transmission of trachoma. This is an interesting observation with potential implications for the control of trachoma in Ethiopia though, as the authors state, it is not clear that the concentration of ivermectin in human faeces is sufficient to inhibit the breeding of M sorbens and further work is required to establish if this is the case. 

The main limitation of this observational study is that this difference may be due to a number of potential confounding factors. The most important of these is the possibility that the prevalence of active trachoma was higher at baseline in the EUs which did not receive ivermectin. This information is available to the authors and should be included in this paper. Other confounders not mentioned in this paper include the number of rounds of azithromycin treatment received in each EU, and altitude, which has been previously shown to be inversely correlated with the prevalence of active trachoma in Ethiopia 1. 

1. Alemayehu W, et al Active trachoma in children in central Ethiopia: association with altitude. Trans R Soc Trop Med Hyg. 2005;99(11):840-3

Reviewer #3: This is an ecological/observational study in trachoma control pointing out that evaluation units (EUs) which also received ivermectin MDA had a better trachoma control outcome in terms of recrudescence/persistence of TF. While this is an interesting observation, and question the hypothesis has been generated by post-hoc consideration of the data. It is not possible to say whether the sample size was adequate or assess power as this depends on how the various competing explanations/confounders would be addressed, although this is likely to be all the data that is available that bears on the question. The primary analysis presented is a (commendably) simple chi-square comparison of whether an EU received IVM versus the outcomes of trachoma recrudescence and persistence - although maps are presented where some potential confounders are overlaid- EUs receiving IVM were different with respect to land cover and population density and are stated to be similar with respect to access to improved water and sanitation, although no data are presented. The authors have not performed any kind of modelling or analysis but are just asking us to eyeball the maps. I am sure they could do better by attempting a more formal statistical analysis and evaluating the role of these and other potential confounders at EU level or possibly at a finer level of detail, if they have access to cluster level data

Technically data 'obtained through the ITI' should respect the principle that all data compiled by the ITI belongs to the contributing Ministries of Health. I am not sure but presumably the same is true of data accessed through ESPEN and so for regulatory purposes there should be a statement that the Ethiopian MoH has sanctioned the accessing of its data for this purpose and takes responsibility for it?

**Results**

-Does the analysis presented match the analysis plan?

-Are the results clearly and completely presented?

-Are the figures (Tables, Images) of sufficient quality for clarity?

Reviewer #1: Yes, mostly, improvement suggested

Reviewer #2: See comments above. In figure 3, which compares the landcover, population density, access to improved sanitation and access to improved water between EUs with and without persistent or recrudescent trachoma, and in figure 2, it is not possible to distinguish between these two groups. These figures should be reformatted to make the difference clear.

Reviewer #3: The lack of an analysis plan, or indeed of an analysis specifically designed to evaluate confounding is a weakness as outlined above

The maps are well presented and the basis for the observations are clear 

It would help to provide EU level data in a supplement so that independent modeling could be done

**Conclusions**

-Are the conclusions supported by the data presented?

-Are the limitations of analysis clearly described?

-Do the authors discuss how these data can be helpful to advance our understanding of the topic under study?

-Is public health relevance addressed?

Reviewer #1: Yes, no (suggestions given), yes, yes

Reviewer #2: The main limitation of this observational study is that this difference may be due to a number of potential confounding factors. The most important of these is the possibility that the prevalence of active trachoma was higher at baseline in the EUs which did not receive ivermectin. This information is available to the authors and should be included in this paper. Other confounders not mentioned in this paper include the number of rounds of azithromycin treatment received in each EU, and altitude, which has been previously shown to be inversely correlated with the prevalence of active trachoma in Ethiopia 1. 

The public health relevance of the observation that districts receiving ivermectin in addition to azithromycin is clear, since there are many districts in Ethiopia in which the prevalence of active trachoma remains high in spite of many rounds of azithromycin mass treatment. As stated in the paper, further research is needed to establish whether the concentration of ivermectin in human faeces is sufficient to inhibit the breeding of M. sorbens

Reviewer #3: No!. In my view the authors have no clear path to implicating an effect on M.sorbens reproductive capacity in the data. Transmission of trachoma by M.sorbens has not so far been demonstrated at scale in Ethiopia (though I am aware of some preliminary data from Oromia). Short term sterilising effects mediated via dung in IVM treated cattle on reproduction of M.nevili (an unrelated species- there are huge differences in behaviour among Muscids) and the observation that there might be fewer maggots in public toilets beg all sorts of questions (cattle receive much more IVM than humans, M sorbens is not considered to breed effectively in latrines and the appearance/disppearance of unspecified maggots utterly non-specific given the diversity of flies documented as emerging from/breeding in public toilets in various settings). The authors do not explain how the IVM MDAs were done, but it would be conventional to exclude under 5s , a group which contributes to open defecation and probably under-utilises public toilets or household latrines, because of a paucity of safety data. So I find the argument that this mechanism is definitively 'suggested' by the data as unduly speculative 

Possible confounders of the observed association need to be properly addressed- although there have rarely been formal studies of transmission there is generally considered to be less trachoma transmission in 'forested' areas than in more arid places and there are probably a wide range of socioeconomic and behavioural factors that differ between them. In general the use of EU level data smoothes out this kind of detail

**Editorial and Data Presentation Modifications?**

Reviewer #1: Thank you for the opportunity to review this extremely interesting paper. An important overall comment is that although this analysis used publicly-available data, they are data that pertain to Ethiopia alone, and it seems wrong that no authors affiliated to Ethiopian institutions – particularly the Ethiopian FMOH, who would have contributed the relevant data to ESPEN – are currently involved. I think that oversight should be rectified.

6: please change to “Trachoma is the most common infectious cause of blindness.” The blindness itself is not infectious.

6: “A disease associated with lack of access to sanitation and low hygiene standards.” This is a sentence fragment: please edit.

8: E stands for “environmental improvement” (not “improvements”)

12, 13, 14, 18 (and elsewhere): onchocerciasis, lymphatic filariasis, ivermectin and albendazole should not be capitalized unless the word concerned starts a sentence.

14: please delete “other”; including it implies that previously mentioned organisms are also insects, which some might interpret as the pathogens responsible for the diseases in the previous sentence

22: “benefit from a higher impact against trachoma” – please change to reflect the actual outcomes studied.

23: please change “among the population” to “MDA”

25: please change “taking the assessment further” to “further work”

28: same comment as in abstract

28: “A disease associated with poverty, which if left untreated leads to damage to the patient’s eye.” This is another sentence fragment. Also: is there only one patient? Is there only one eye? Is the person or people involved a patient or patients before they seek care?

29: I think the authors might mean “culminates” rather than “cumulates”

30: “Trachoma is strongly associated with lack of access to sanitation”. I think this statement should be referenced. There are lots of small studies that demonstrate the association, but I think the largest (and most convincing) are 1. Garn JV, Boisson S, Willis R, et al. Sanitation and water supply coverage thresholds associated with active trachoma: modeling cross-sectional data from 13 countries. PLoS Negl Trop Dis 2018; 12(1): e0006110. and 2. Sullivan KM, Harding-Esch EM, Keil AP, et al. Exploring water, sanitation, and hygiene coverage targets for reaching and sustaining trachoma elimination: G-computation analysis. PLoS Negl Trop Dis 2023; 17(2): e0011103.

31: suggest delete the words “coupled with low hygiene standards” – this could be interpreted as sounding pejorative or judgmental.

33: please change “bacteria” to “bacterium”

43: please change “cease” to “correct”

47: please change “improvements” to “improvement”

55: please delete “Sightsavers” here. The particular funder should have no bearing on interpreting the weight of the research; including it here is likely to be viewed as an advertorial

58: “even” in this sentence sounds a bit breathless. Please edit

60: a prior trial (and cleaner study, in the sense that only the two relevant agents were administered) for coadministration of ivermectin and azithromycin is: 1. Romani L, Marks M, Sokana O, et al. Feasibility and safety of mass drug coadministration with azithromycin and ivermectin for the control of neglected tropical diseases: a single-arm intervention trial. Lancet Glob Health 2018; 6(10): e1132-e8.

63: the abbreviation “TF” has not previously been defined in this manuscript. Persistence and recrudescence of TF have similarly not been previously defined.

65: Ethiopia has not only the highest burden of trachoma, but the greatest recorded incidence to date of persistent and recrudescent TF

68: “ratified” is far too strong here. The definitions are working definitions proposed at a WHO informal consultation. Incidentally, when abbreviated, “WHO” does not take the definite article. 

69-70: the phrasing here conflates “trachoma” and “TF”. One is a disease; the other is a specific sign

72: suggest delete the words “the data processing and”

77: the analysis is secondary. The data are not secondary.

89: presumably there were some EUs that did not have persistent TF or recrudescent TF? (As a grammatical aside here, it is not the EUs that are “persistent or recrudescent” – it’s TF in those EUs.)

91: having defined the abbreviation “EU”, it should be used consistently

101: “To examine the influence of these variables on this investigation available geographic data have been collated for climate (21), landcover (22), population density (23), sanitation and water access (24). These data were overlaid with the plots of EUs within our datasets for examination.” This is important: it's trying to address an important confounder that might explain differences between EUs co-endemic for oncho or LF and those that have only required interventions against trachoma. Details of the way in which this was built into the analysis are necessary; this would ideally be more sophisticated than simply overlaid visualisations.

106: “EUs that could not be aligned from ESPEN into GeoConnect were discounted from our dataset prior to assessment commencing”. I may have misunderstood what this actually means, but feel that the number “discounted” should be included in the flow diagram that forms Figure 1.

121: “During the process of merging GeoConnect and ESPEN datasets, 10 GeoConnect IDs from the trachoma dataset matched two ESPEN IDs. To resolve this, one of the ESPEN ID was randomly retained as a match for each GeoConnect ID.” This seems like a rather arbitrary way to resolve the problem. Can a more considered investigative approach be explored?

124: “Among the 208 EUs retained for trachoma analyses, 28 had received IVM MDA, starting on or after the year of trachoma baseline survey and occurring on or before the last year of survey on trachoma.” The requirement for IVM MDA to commence *after* (or at worst, in the same year as) the baseline trachoma survey seems sensible to appropriately examine differential reductions in prevalence. But this requirement was not apparent in the methods section. It should be.

Tables 1 and 2: Given the amount of data available, use of three significant digits for the percentages provided here is excessive. Please round to whole numbers.

131 onwards: please don’t describe the *EUs* as being “persistent” or “recrudescent”. These labels can and should be applied to the situation of active trachoma in those EUs, but not the EUs themselves.

143: please delete the adjective “continued” here – it’s redundant.

Figure 2 is surprisingly hard to read at a glance, which should be the role of this type of figure. Can the authors please experiment with other ways of displaying the categories to see if they can construct a more digestible visualization?

149: “Showing rainfall patterns are not vastly different between the EUs receiving IVM MDA compared to trachoma EUs which met the inclusion criteria for this study.” This is a sentence fragment. Please edit.

The analysis of these data and the data shown in Figure 3 is a bit basic; it assumes that any relationship would be visually obvious and ignores possible interactions of the known confounders identified. Suggest further consultation with a statistician.

170: risk of *active* trachoma

179: “The plasma half-life of IVM in humans being approximately 18 hours following oral administration.” This is a sentence fragment: please edit.

181: “This anticipated active lifespan is validated by the effect on gastrointestinal nematodes ceasing 12 days post dosing in humans.” A reference should probably be provided to support this statement, but even if it is evidenced, I am not sure that it is relevant. The concentrations needed to produce “effect” on nematodes in the lumen or attached to the mucosa of the GI tract are not necessarily the same as those needed to affect viability of flies ovipositing on soil-exposed faeces.

183: “Initial interest focuses on IVM treatment in humans to deliver M. sorbens control, with complimentary laboratory investigation into veterinarian IVM treatment of domestic livestock.

Investigation into use in livestock will guide any addition as further support, added in certain field

settings if human IVM treatment is not achieving M. sorbens control (26).” This sounds premature, given that it is not yet proven as an intervention, but also misguided, because (as the authors point out in the next sentence), “Evidence from Gambia shows M. sorbens population responsible for trachoma transmission reproduce only in human excreta (5).” Ref 26 is an observational study that is not in any way convincing, and should not be included in a serious discussion of the evidence. The rest of this paragraph is highly speculative and should be deleted or revised.

A paragraph giving serious considerations to the limitations of the study would augment the paper. Some of the limitations are alluded to but should be identified explicitly as limitations, and gathered together in the usual style of scientific papers.

Reviewer #2: A number of points need to be clarified in this manuscript. TF 1-9 is not defined. The differences between an impact survey and a surveillance survey, and between a persistent and a recrudescent EU are not clear. The authors state that trachoma is the leading cause of infectious blindness, but blindness is not infectious. It is the leading infectious cause of blindness. They state that evidence from Gambia shows the M. sorbens population responsible for trachoma transmission reproduces only in human excreta. This is incorrect. The paper they cite states that “Musca sorbens emerged from human (6/9 trials), calf (3/9), cow (3/9), dog (2/9) and goat (1/9) faeces”.

Reviewer #3: I am OK with the way they have presented the data but I consider that the analysis and discussion are primary weaknesses

The maps are great but not adequate to enable the results to be evaluated independently- could a dataset of all EU level variables examined be appended?

**Summary and General Comments**

Reviewer #1: (No Response)

Reviewer #2: The fact that evaluation units (EUs) which received mass treatment with ivermectin in addition to azithromycin were less likely to have persistent or recrudescent active trachoma than those which received azithromycin alone is an interesting observation of potential public health relevance. The main limitation of this observational study is that this difference may be due to a number of potential confounding factors. The most important of these is the possibility that the prevalence of active trachoma was higher at baseline in the EUs which did not receive ivermectin. This information is available to the authors and should be included in this paper. Other confounders not mentioned in this paper include the number of rounds of azithromycin treatment received in each EU, and altitude, which has been previously shown to be inversely correlated with the prevalence of active trachoma in Ethiopia 1. 

Given the fact that the data included in this study were all collected in Ethiopia, it is surprising that none of the authors appears to be from that country.

Reviewer #3: The observation that on the face of it MDA with IVM may be associated with improved trachoma control outomes at EU level is novel and potentially important. However there are a wide range of potential confounders and missed analytical opportunities and evidence for the speculated mechansism is severely lacking:

1) is there evidence that M.sorbens transmits trachoma in any of the EUs?

2) is there evidence that there is less transmission of trachoma underlying the reduced risk of labelling an EU as persistent or recrudescent (tests for infection?)

3) is there evidence that any other parameters of fly borne transmission (e.g face-fly contact) or the observations of flies attacking ocular/nasal secretions that may be reduced following IVM MDA? 

4) It would normally be considered that flies are quite successful at reproducing (they breed like flies after all) and there is very little evidence that IVM affects this particular fly to the extent of temporarily altering its behaviour(normally considered to be quite plastic) in ways that would affect its reproductive success for more than a few days

5) Toxicity to egg laying female flies may have more impact than toxicity to maggots (400+ produced each time) 

6) Is there any association of TF outcome with IVM coverage? 

7) of the very large number of potential confounders can any of them be examined at cluster level? 

8) what is the effect of lumping the data together at EU level and is a sensitivity analysis at a subunit level informative 

9) under 5s are at high risk of trachoma but are not usually treated with IVM. Is the availability of a reservoir of faecal material in all environments which has not come from an IVM treated individual not a concern for the hypothesis? 

I'd favour representing this information with a more determined analytical approach to confounding and less speculative conclusions. There is not really any evidence so far that this is anything to do with the reproductive capacity of M.sorbens, which, if anything, is known to be quite robust. The observation itself is interesting and potentially important- it would merit a more careful examination of all the potential explanations both analytically and in discussion

PLOS authors have the option to publish the peer review history of their article (what does this mean?). If published, this will include your full peer review and any attached files.

Reviewer #1: No

Reviewer #2: Yes: David Mabey

Reviewer #3: No
---

## [Decision Letter · Decision Letter 1]

11 Aug 2023

Dear Dr Selby,

Thank you very much for submitting your manuscript "Potential mitigating role of ivermectin on the spread of Chlamydia trachomatis by Musca sorbens." for consideration at PLOS Neglected Tropical Diseases. As with all papers reviewed by the journal, your manuscript was reviewed by members of the editorial board and by several independent reviewers. The reviewers appreciated the attention to an important topic. Based on the reviews, we are likely to accept this manuscript for publication, providing that you modify the manuscript according to the review recommendations. 

Dear authors

Please address the existing comments - all minor! - and resubmit, so that it can be considered accepted.

Sincerely,

Susana Vaz Nery

Academic Editor

Álvaro Acosta-Serrano

Section Editor

Dear authors

Please address the existing comments - all minor! - and resubmit, so that it can be considered accepted.

Reviewer's Responses to Questions

**Key Review Criteria Required for Acceptance?**

**Methods**

-Are the objectives of the study clearly articulated with a clear testable hypothesis stated?

-Is the study design appropriate to address the stated objectives?

-Is the population clearly described and appropriate for the hypothesis being tested?

-Is the sample size sufficient to ensure adequate power to address the hypothesis being tested?

-Were correct statistical analysis used to support conclusions?

-Are there concerns about ethical or regulatory requirements being met?

Reviewer #1: Yes, yes, yes, dictated by available data, probably, no

Reviewer #2: yes to all

**Results**

-Does the analysis presented match the analysis plan?

-Are the results clearly and completely presented?

-Are the figures (Tables, Images) of sufficient quality for clarity?

Reviewer #1: Yes, yes, yes

Reviewer #2: yes to all

**Conclusions**

-Are the conclusions supported by the data presented?

-Are the limitations of analysis clearly described?

-Do the authors discuss how these data can be helpful to advance our understanding of the topic under study?

-Is public health relevance addressed?

Reviewer #1: Partially (see comments), yes (the authors are overly sceptical of their own work), yes, yes

Reviewer #2: yes to all

**Editorial and Data Presentation Modifications?**

Reviewer #1: Purely from a stylistic perspective, it sounds a bit mechanical to have the first three sentences in the abstract all start with “Trachoma is…”

21: Suggest change “and whether co-administration could have controlled the putative vector, unintentionally enhancing trachoma control.” to “and whether co-administration was associated with greater likelihood of trachoma control”. With this kind of retrospective study, one cannot infer causation – and no data are presented on vector control per se. Vector control is just a hypoththesis to explain the observed association.

24-25: suggest delete “have only”

48. “Which if left untreated leads to damage to the infected person’s eyes.” – this is a sentence fragment.

53: please delete “shared” in front of “fomites”

57-58: suggest delete “to complete their life cycle”. With the sentence structure as it currently stands, it sounds like the authors are describing the life cycle of human faeces. But in any case, the notion that flies need to reproduce in order to complete their life cycle should be redundant in a scientific journal in 2023.

63: Please add “from trachoma” after “Blindness”

64: Please delete the word “on” after “encompass”

65: please change “to stop blinding trachoma” to “to eliminate trachoma as a public health problem”. “Blinding trachoma” sounds like you’re swearing in the 1920s.

73: the meaning of “other” in front of “flies” here is unclear. It could be taken to imply that the authors regard dung breeding beetles as “flies”. Would this sentence still be accurate if changed to read, “Reproductive impact of IVM has also been recorded for dung breeding beetles”?

82: please use only lower-case letters, and an em-dash without flanking spaces, for “trachomatous inflammation—follicular”

98: please delete the phrase “sourced from the International Trachoma Initiative” – the data come from the Ethiopian government.

109 and 111: the EUs are not persistent or recrudescent. The EUs have persistent or recrudescent active trachoma. 

109: Having defined the abbreviation TF previously, please use it consistently. (TF does not represent “Trachoma Follicular” – it represents “trachomatous inflammation—follicular”.)

109: the word “prevalence” should appear in here

134: “used to describe precipitation, elevation, Univariate logistic models were used to explore” should probably read, “used to describe precipitation and elevation. Univariate logistic models were used to explore”

137-138: “A backward selection procedure was used to retain the candidate model with the best subset of variables.” The word “best” here is vague. Do the authors mean “with the subset of variables that demonstrated statistically significant associations with the outcome of interest, using an alpha of 0.05”?

139: I am not sure what “and numerical issues” means. Should these words be deleted? Or do the authors want to say that “a history of receiving ivermectin completely predicted an absence of active trachoma recrudescence”?

228: Suggest delete “In a multivariable model with all variables entered simultaneously, the association between IVM and persistence status is no longer statistically significant (p=0.15).” and change the opening of the subsequent sentence to read “Using the backward selection process, the best candidate multivariable model…”. The final multivariable model is the one that readers are interested in, not all possible candidate models.

237: Similarly, suggest delete the material undermining the hypothesis that reads “The results are not clear-cut as association becomes non-significant.” If you are in any doubt about how to undertake and interpret the multivariable modelling, I strongly recommend enlisting additional advice from an experienced modeller!

247: “Delivering an evaluation on the repurposing of IVM to support trachoma campaigns in difficult foci.” This is a sentence fragment.

255: Please please please please please delete this whole paragraph and do not build a hypothesis about treating animals. The De Sole paper cited has been hugely overinterpreted; it is not the basis for intervention. Emerson showed quite clearly that the subpopulations of M. sorbens that oviposit on human faeces go to human eyes; M. sorbens that oviposit on cattle dung go to cows’ eyes. They are distinct ecological niches. This paragraph builds unnecessary speculation on a good hypothesis, does so on what I believe to be erroneous scientific grounds, and is therefore likely to be misleading.

291: Please delete “but the results were not clear-cut”.

295: I strongly suggest deleting “Field research should consider the introduction of IVM MDA in selected case EUs with both recrudescent or persistent trachoma. Along with control EUs with recrudescent or persistent trachoma where IVM MDA is not introduced. In each selected EU, SAFE strategy including AZM MDA should continue as currently managed. Investigation should then longitudinally measure M. sorbens relative densities along with trachoma prevalence to measure force of infection in each EU.” I think this is premature. 

305: “We lack the granularity of individual-level data”. I don’t think individual-level data would be very helpful – prevalence of TF is the outcome of interest, and population-level effects are likely to be far more important for the hypothesized mechanism of impact on fly reproduction.

Please check the formatting of the references. Ref 23 appears as if it had been written by Dr W. H. Organization. (The fix is to put a comma after Organization in the author field of the relevant record in your reference management software.) Refs 22 and 32 have the same problem.

Reviewer #2: The authors have satisfactorily addressed all the comments raised by the reviewers

**Summary and General Comments**

Reviewer #1: Thank you for the opportunity to re-read. This is a strong study on an important topic. The additional analyses incorporated since the original submission defintely augment the work. However, I would strongly recommend asking an experienced modeller to read the paper and offer input before resubmission, either as a co-author or reviewer. I am not such a person, but know enough to feel that the authors are still not quite striking the right tone in the way they have approached the analysis and interpretation, and this study is sufficiently novel and important that getting this all right is something in which i think it is worth investing time. I am sorry to make this suggestion and therefore probably slow things down, because the authors are undoubtedly anxious to put this in the public domain as soon as possible, as they should be.

Reviewer #2: The authors have satisfactorily addressed all the comments raised by the reviewers

PLOS authors have the option to publish the peer review history of their article (what does this mean?). If published, this will include your full peer review and any attached files.

Reviewer #1: No

Reviewer #2: Yes: David Mabey

Figure Files:

Data Requirements:

Reproducibility:

References

---

## [Editor Report · Decision Letter 2]

13 Sep 2023

Dear Dr Selby,

We are pleased to inform you that your manuscript 'Potential mitigating role of ivermectin on the spread of Chlamydia trachomatis by Musca sorbens.' has been provisionally accepted for publication in PLOS Neglected Tropical Diseases.

Best regards,

Susana Vaz Nery

Academic Editor

Álvaro Acosta-Serrano

Section Editor

---

## [Editor Report · Acceptance letter]

9 Oct 2023

Dear Dr Selby,

We are delighted to inform you that your manuscript, "Potential mitigating role of ivermectin on the spread of *Chlamydia trachomatis* by *Musca sorbens*.," has been formally accepted for publication in PLOS Neglected Tropical Diseases.

Best regards,

Shaden Kamhawi

co-Editor-in-Chief

Paul Brindley

co-Editor-in-Chief
